# Tackling Imbalanced Class in Federated Learning via Class Distribution Estimation

## Abstract

Federated Learning (FL) has become an upsurging machine learning method due to its applicability in large-scale distributed system and its privacy-preserving property. However, in real-world applications, the presence of class imbalance issue, especially the mismatch between local and global class distribution, greatly degrades the performance of FL. Moreover, due to the privacy constrain, the class distribution information of clients can not be accessed directly. To tackle class imbalance issue under FL setting, a novel algorithm, FedRE, is proposed in this paper. We propose a new class distribution estimation method for the FedRE algorithm, which requires no extra client data information and thus has no privacy concern. Both experimental results and theoretical analysis are provided to support the validity of our distribution estimation method. The proposed algorithm is verified with several experiment, including different datasets with the presence of class imbalance and local-global distribution mismatch. The experimental results show that FedRE is effective and it outperforms other related methods in terms of both overall and minority class classification accuracy.

## 1 Introduction

Federated Learning (FL) was first proposed (McMahan et al., 2017) when they were developing the application of next-word prediction on mobile keyboard. It enables multiple clients to collaboratively learn a machine learning model, without sharing their locally stored raw data (Li et al., 2020a). This property greatly reduce the communication cost and preserve the privacy of clients, which makes FL become an upsurging research direction, not only in machine learning community but also in a variety of engineering applications, including communication (Wang et al., 2022; Niknam et al., 2020; Mills et al., 2019), edge computing (Zhang et al., 2021a; Wang et al., 2019a;b), and energy engineering (Saputra et al., 2019; Hamdi et al., 2021; Cheng et al., 2022).

Standard FL consists of four major steps, which are client selection, broadcast, client computation, and aggregation (Kairouz et al., 2021). In each iteration, the central server will select a subset of clients in each global iteration at first, and then broadcast the global model to them. After receiving the global model, the selected clients will perform model update based on their local dataset given the global model as initial condition, and then upload updates to the server. As the final step, the server will aggregate the collected information to update the global model, and then start a new iteration.

In FL framework, one of the most difficult challenges is class imbalance issue. Class imbalance means that the data distribution among all classes are not uniform. In other words, the majority of data samples may belong to certain classes, and other minority classes may only have a small amount of data. Class imbalance results in low classification accuracy on minority classes, and also slow down the training speed. In the literature, several methods have been proposed to resolve class imbalance issue in centralized machine learning scheme. In general, the methods can be categorized as data-level methods, algorithm level methods, and hybrid methods (Johnson & Khoshgoftaar, 2019). For data-level method, Jo & Japkowicz (2004) proposed a cluster-based sampling scheme to tackle the class imbalance issue. For algorithm level method, Ling & Sheng (2008) proposed the cost-sensitive learning to improve the classification performance on minority class. For hybrid method, Sun et al. (2007) integrated both sampling techniques and cost-sensitive learning and showed a significant performance boost in most of the cases. However, in FL, since all training data

are distributed and stored locally and not exchangeable, it is infeasible to apply data-level methods. Besides, due to the inconsistency of local and global class imbalance, algorithm-level methods such as cost-sensitive learning are not effective and may even impose negative effect on the performance of global model (Wang et al., 2021). Thus, those methods that have shown great achievements on class imbalance issue in centralized machine learning cannot be applied directly in FL, and new algorithm has to be considered under the constrains of FL scheme.

In this work, the class imbalance issue in FL is addressed, especially the issues of global class imbalance and mismatch between local and global class distribution. To tackle the challenges mentioned previously while preserving the privacy condition, a new FL algorithm is proposed. Due to the privacy condition, the local dataset and local class distribution of each clients are not accessible. Thus, an estimation method to estimate the class distribution is developed, requiring no extra client dataset information. Based on the estimated class distribution, the loss re-weight method can be applied to handle the global class imbalance. The proposed method requires no additional client information, so the privacy safety can be guaranteed. Besides, from the experimental results, the proposed method has shown a significant improvement on the classification accuracy of minority class under the class imbalance scenario.

**Contribution.** In summary, the contributions of this work is as follows.

1. The proposed estimation method can estimate the class distribution without additional client information, so the privacy safety can still be guaranteed. Moreover, the estimation method doesn't require much extra computation, so the overall efficiency of FL algorithm is not degraded. The theoretical analysis is also provided to support the validity of the proposed distribution estimation method.

2. The proposed new FL algorithm based on class distribution estimation achieves significant improvement on handling the class imbalance issue in FL, and the proposed algorithm is verified by experiments with different heterogeneity level and different dataset.

## 2 RELATED WORKS

### 2.1 FEDERATED LEARNING

Due to the development of edge devices and the increasing popularity of mobile devices, the distributed machine learning has become one of the most popular direction of machine learning research. However, common mobile devices such as smart phone and wearable electronics have limited computation and communication power. Moreover, those mobile devices contains personal information which is private and not exchangeable. Thus, a new distributed machine learning problem, also known as Federated Learning (FL), has to be considered to overcome the challenges of communication and computation cost, data heterogeneity, and privacy. To overcome the novel challenges, FedAvg (McMahan et al., 2017) was first proposed and addressing the communication efficiency problem. Since then, many researches have been conducted to resolve challenges in FL. Li et al. (2020b) proposed the algorithm FedProx to tackle the data heterogeneity issue by introducing a proximal term in local objective function, and provided theoretical analysis on the convergence. To improve the convergence speed and reduce the communication cost, in Reddi et al. (2020) different adaptive learning techniques were applied in the aggregation steps. Wang et al. (2020b) investigated the fundamental cause of heterogeneity, and proposed a normalized averaging algorithm FedNova to improve the FL performance. Li et al. (2021) applied local batch normalization to tackle the feature-shift non-iid issue, which is another type of data heterogeneity. Some recent works in literature also provided improvement on FL algorithm by proposing different client selection scheme (Nishio & Yonetani, 2019; Ribero & Vikalo, 2020; Balakrishnan et al., 2021).

### 2.2 CLASS IMBALANCE IN FEDERATED LEARNING

One of the most difficult yet important challenges in FL is class imbalance. The class imbalance issue can be categorized as local imbalance and global imbalance (Wang et al., 2021). Works mentioned previously were addressing on data heterogeneity, which belongs to local imbalance. However, as stated in section 1, due to the mismatch between local and global distribution, only handling local data imbalance is not enough. To deal with class imbalance issue, the algorithm Astraea

from Duan et al. (2019) introduced mediators between local clients and central server to facilitate scheduling and aggregation. The proposed method requires real client data distribution, which violates the privacy condition. Furthermore, if the distribution information are leaked during the transmission, malicious attackers who intercept such information could analyze client preference and launch targeted attacks (Wang et al., 2021). To preserve client privacy, some previous methods in literature build proxies to estimate the class distribution. Yang et al. (2021) suggested that the gradient magnitude can be proxy to estimate the class distributions of clients, and utilized it to design a new client selection method. Wang et al. (2021) suggested that there exists proportional relation between gradient magnitudes and sample quantities. Based on this argument, Wang et al. (2021) developed a monitoring scheme that can estimate the class distribution, and introduced the algorithm, Ratio-Loss, to perform class-wise re-weight. However, the suggested proportional relation between gradient magnitude and sample quantity is based on strong assumptions, which may not be feasible in practice. As another branch of research direction, there are works utilize active learning and reinforcement learning to eliminate the effects of class imbalance (Goetz et al., 2019; Zhang et al., 2021b; Wang et al., 2020a). However, these methods usually rely on different client selection schemes. In practical setting of FL, available clients in each communication round are different and uncontrollable by server. Therefore, methods applying client selection have limited applicability (Shen et al., 2021). Compared with others works, we propose a new FL algorithm and a new class distribution estimation method that can facilitate the proposed algorithm without violation of privacy condition. Besides, the proposed algorithm does not rely on active client selection schemes. Thus, it has less limitation on applicability in practice.

## 3 METHODOLOGY

### 3.1 DEFINITION AND BACKGROUND

In this paper, the multi-class classification problem in FL is considered. Let $\boldsymbol{x} \in \mathbb{R}^d$ be the $d$ dimensional input and $\boldsymbol{t} \in \mathbb{R}^Q$ be the target one-hot vector of $Q$ classes. Under the FL framework, each client $k$ has a local objective $F_k(\boldsymbol{\omega}) = \frac{1}{n_k} \sum_{j=1}^{n_k} l_j(\boldsymbol{\omega}, \boldsymbol{x_j}, \boldsymbol{t_j})$, where $n_k$ is the number of samples of client $k$, $\boldsymbol{\omega}$ is the model parameters, and $l$ is the loss function. In FL, the goal is to minimize the global average of loss functions, which can be stated as follows:

$$\min_{\boldsymbol{\omega}} F(\boldsymbol{\omega}) := \sum_{k=1}^{K} p_k F_k(\boldsymbol{\omega}) \tag{1}$$

where $K$ is the total number of clients, $p_k = \frac{n_k}{n}$, and $n = \sum n_k$ is the total number of samples. The challenge of class imbalance in FL is that for each client $k$, their local class distribution $\boldsymbol{D}_k$ are different. Besides, there may be a huge mismatch between local and global class distribution, that is, $\boldsymbol{D}_k$ and global class distribution $\boldsymbol{D}_g$ can be very different. Moreover, due to the privacy assumption in FL, local data and data distribution can not be accessed directly. To tackle these challenges, a new FL algorithm based on class distribution estimation is developed, and will be introduced in detail in the following sections.

### 3.2 DISTRIBUTION ESTIMATION

As discussed in previous section, the biggest challenge is that the distribution information is unknown. If those information can be estimated without violation of privacy, one can utilize those information to handle the class imbalance problem. Therefore, the problem lies in how to estimate the class distribution under the framework of FL. The class distribution estimation method developed in this work is based on the following observation:

**Observation**: When updating the model parameters $\boldsymbol{\omega}$ by Gradient Descent (GD) or Stochastic Gradient Descent (SGD), in the parameter space $\boldsymbol{\omega}$ will converge to a flat region in the early stage of training, such that the prediction output of all input from the model is approximately equals to the class distribution ratios.

The observation can be demonstrated in the following experiment on synthetic data. Consider a binary classification problem, where the sample points from two classes $C_1$ and $C_0$ are distributed on the 2-D plane as shown in Fig.1. Without loss of generality, assume that the minority class $C_0$

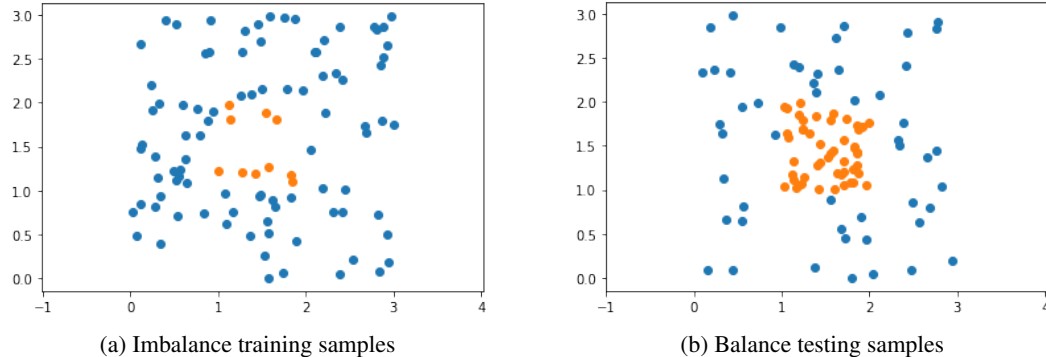

(a) Imbalance training samples          (b) Balance testing samples

Figure 1: 2-D synthetic data samples for the binary classification problem. Blue dots represent samples from majority class $C_1$ and orange dots represent samples from minority class $C_0$. (a) Imbalance training samples with $n_0 = 10$ and $n_1 = 90$. (b) Balance testing samples with $n_0 = n_1 = 50$. $C_0$ is generated by uniformly random sampling such that $x = \{[x1, x2] : x1, x2 \in (1, 2)\}$, and $C_1$ is generated by uniformly random sampling such that $x = \{[x1, x2] : x1, x2 \in (0, 3) \setminus (1, 2)\}$.

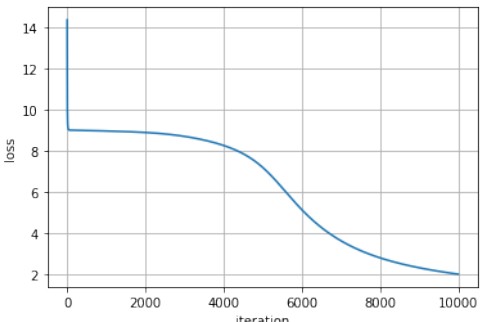

Figure 2: Training loss with respect to training iteration for the binary classification problem on imbalance synthetic data. The plateau region can be observe at the early stage of training (before 2000 iteration). In this region, the output values are approximately equals to 0.9

(labeled as target 0) and majority class $C_1$ (labeled as target 1) have samples quantities $c_0$ and $c_1$, respectively, and the imbalance level $\rho = c_1/c_0 = 9$. To solve this classification problem, a simple one-layer fully connected neural network with 4 hidden Sigmoid neurons is used and the square loss function is applied. The prediction output is the probability of input belonging to $C1$. The plot of training loss with respect to iterations on the imbalanced dataset is shown in Fig.2. One can observe that the training loss drops quickly at the beginning and then reaches a plateau. If we look at the prediction output at this region as shown in Table 1, one can observe that the prediction output from both $C_1$ and $C_0$ are approximately equals to $0.9 = n_1/N$, which is the portion of $C_1$ among total number of samples $N$. This phenomenon can also be observed using Binary Cross Entropy (BCE) loss, or by different imbalance level $\rho$. Moreover, the phenomenon can be observed in the multi-class classification problem as shown in Fig.3. The following Theorem 1 explains the cause of this phenomenon from a theoretical point of view.

| 1 | 2 | 3 | 4 | 5 | 6 | 7 | 8 | 9 | 10 |
|---|---|---|---|---|---|---|---|---|----|
| 0.9066 | 0.8865 | 0.9075 | 0.8999 | 0.8865 | 0.8981 | 0.8926 | 0.8921 | 0.8953 | 0.8909 |

Table 1: Ten outputs of samples from synthetic testing dataset in the binary classification problem. Outputs 1 to 5 are from samples in majority class $C_1$ and outputs 6 to 10 are from samples in minority class $C_0$

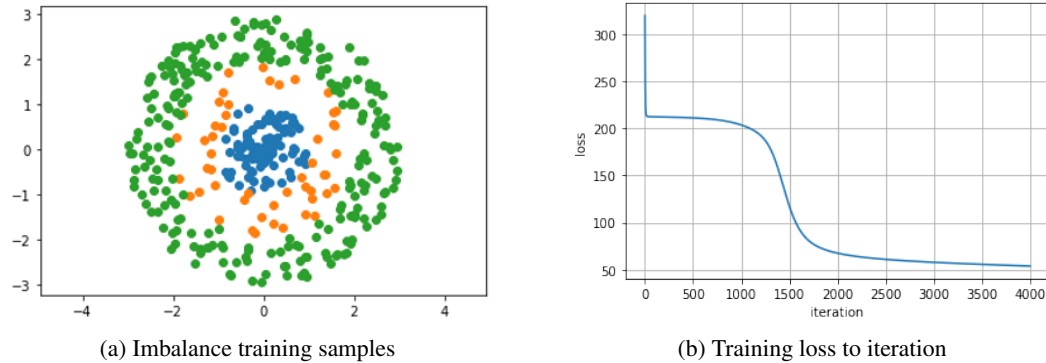

(a) Imbalance training samples        (b) Training loss to iteration

Figure 3: Experiment on synthetic data for the multi-class classification problem. (a) Imbalance training samples from 3 classes with number of samples $[n_1, n_2, n_3] = [100, 50, 250]$ and class distribution $\boldsymbol{D} = [0.25, 0.125, 0.625]$, respectively. (b) The plot of training loss with respect to iteration. At the plateau region, the average output from testing set is $[0.2545, 0.1316, 0.6199] \sim \boldsymbol{D}$ with error percentage less than $6\%$.

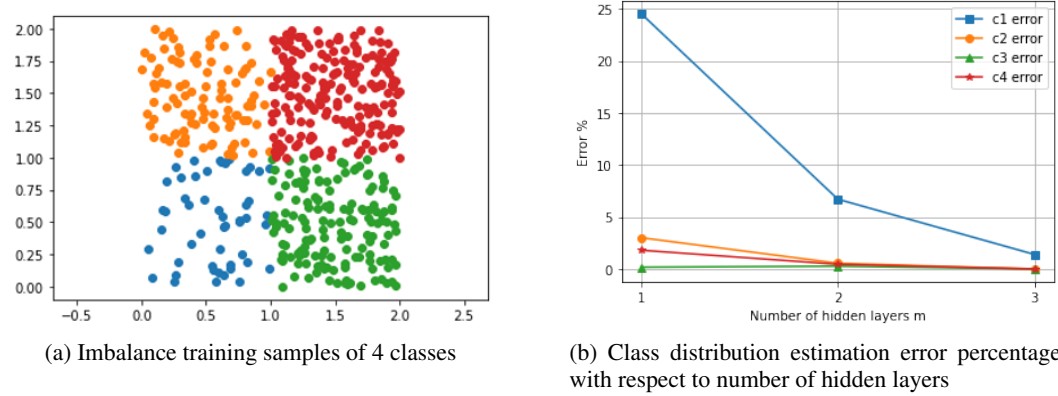

(a) Imbalance training samples of 4 classes      (b) Class distribution estimation error percentage with respect to number of hidden layers

Figure 4: Experiment on synthetic data for the multi-class classification problem with nonequal class sample averages. (a) Imbalance training samples from 4 classes with number of samples $[n_1, n_2, n_3, , n_4] = [50, 100, 150, 200]$ and class distribution $\boldsymbol{D} = [0.1, 0.2, 0.3, 0.4]$, respectively. (b) the distribution estimation error percentages of 4 classes with respect to number of hidden layers. One can observer that the estimation error decays with layers number increased and errors of $c2 - c4$ reach 0 at $m = 3$.

**Theorem 1** *For an $m$-layer fully connected neural network with $d$-dimensional input and $Q$-dimensional output, assume the Sigmoid activation functions and square loss are used, and the training data distribution $\boldsymbol{D} = [c_1/n, ..., c_Q/n]$. In the parameter space, if at some point such that all the model prediction outputs $\boldsymbol{y} = \boldsymbol{D}$, that point is a critical point by first order optimality.*

The proof of Theorem 1 is provided in Appendix, and the proof can also be directly extended to BCE loss case. Theorem 1 explains why such point in the parameter space is preferred during the model updating process, which also provides an explanation of the cause of such phenomenon in a theoretical point of view. Note that even though in the proof it utilizes the assumption that the average of inputs from all classes are the same, in practice the error caused by the violation of this assumption can be eliminated by increasing the number of layers $m$, which is demonstrated in the experiment on synthetic data shown in Fig.4. This result supports the validity of the estimation in practice.

Based on this observation, the class distribution estimation method is designed and summarized in Algorithm 1. At line 2 the $modelUpdate$ function is a commom machine learning process, given

---

**Algorithm 1** Class Distribution Estimation

---

**Input:** minibatch size $B$, number of epochs $E$, learning rate $\eta$, imbalance dataset $\mathcal{D}$, balance auxiliary dataset $\mathcal{A}$, loss function $l$, optimizer $opt = $ SGD

**Output:** estimated class distribution $\hat{D}$

 1: initialize $\boldsymbol{\omega}^0$
 2: $\boldsymbol{\omega}^E \leftarrow modelUpdate(\boldsymbol{\omega^0}, B, E, \eta, \mathcal{D}, l, opt)$
 3: $\hat{D} \leftarrow \frac{1}{|\mathcal{A}|} \sum_{i=1}^{|\mathcal{A}|} f(\boldsymbol{\omega^E}, \boldsymbol{x}_i)$

---

loss function $l$ and the optimizer. In default, the SGD is used because it is more feasible than GD in practice. The hyper-parameters $B, E$ and $\eta$ for SGD are chosen such that the model parameters can reach the plateau region. Note that the range of plateau region is in general large enough as shown previously in experiments, so in practice the hyper-parameters can be chosen without carefully fine-tuning. Some other diagnostic approach (Chee & Toulis, 2018; Bottou et al., 2018) can also be adopted to setup the stopping criteria of $modelUpdate$, which is beyond the scope of the discussion here. In line 3 of Algorithm 1, after obtaining the updated model, a balance auxiliary dataset $\mathcal{A}$ is used to estimate the class distribution. As discussed previously, the prediction outputs from any inputs are approximately equal to the class distribution during the early stage of training. By this property we calculate the average output values from $\mathcal{A}$ as the estimated class distribution $\hat{D}$. $f(\boldsymbol{\omega}, \boldsymbol{x})$ here denotes the prediction output of the neural network given model parameters $\boldsymbol{\omega}$ and input $\boldsymbol{x}$, and $|\cdot|$ operator denotes the size of the set. In practice, $\mathcal{A}$ can be just a small portion of data from public dataset and stored in the central server, so it step will not incur much extra computational cost.

## 3.3 OUR METHOD

The flow chart of the proposed new FL method addressing on class imbalance is presented in Fig.5, and the detail of the algorithm is summarized in Algorithm 2. In order to incorporate the class distribution estimation stated in Algorithm 1 into FL framework, we have to train the model locally, and utilize the locally updated model to estimate class distribution in the server. In the first global iteration, server distributes initial model $\boldsymbol{\omega}^0$ to all clients, and each client trains the model on their local dataset $\mathcal{D}_k$ and obtains updated model $\boldsymbol{\omega}_k$, similar to line 2 in Algorithm 1. Afterward, updated local models are sent back to the server, and the server can use an auxiliary dataset $\mathcal{A}$ to estimate $\hat{D}_k$ for all clients $k$. Note that the hyper-parameters in $modelUpdate$ at line 5 can be tuned separately from those at line 15. In this work, the square loss function $l = (\boldsymbol{y} - \boldsymbol{t})^2$ is used, whose validity is verified experimentally and theoretically in previous section.

With the estimated class distributions, the global class imbalance issue can be tackled my applying loss re-weight method, where the loss weight vector $\boldsymbol{R} = [R_1, ..., R_Q] \in \mathbb{R}^Q$ is calculated at line 9. The function $getLossReweight()$ is defined in Algorithm 2. The idea is that the global class distribution $\boldsymbol{D}_g$ can be estimated based on the estimated client class distribution $\hat{D}_k$, client sample amount $n_k$, and total sample amount $n = \sum n_k$. The loss weight for each class is assigned inverse proportionally to the corresponding global class ratio, so the minority class will receive higher weight. In the function $getLossReweight()$, $\alpha$ and $\beta$ are hyper-parameters that can be tuned. In this work, $\alpha = 1$ and $\beta = 0.01$ are used to achieve the best overall performance.

The second half of Algorithm 2 is the standard FL process. In each global iteration, $CK$ clients are randomly selected, where $C \in (0, 1]$ is the portion of client selection. The selected clients will perform local update, and the sever will then aggregate updated local models to obtain a new global model. Note that the loss function used in local model update is a weighted loss function $L$ depending on the weight vector $\boldsymbol{R}$. In this work, the weighted Cross Entropy Loss is used as stated in the following:

$$L(\boldsymbol{y}, \boldsymbol{t}; \boldsymbol{R}) = -\sum_{c=1}^{Q} R_c \log \frac{\exp(y_c)}{\sum_{i=1}^{Q} \exp(y_i)} t_c, \tag{2}$$

where $\boldsymbol{y} = [y_1, ..., y_Q]$ is the prediction probability vector and $\boldsymbol{t}$ is the target one-hot vector.

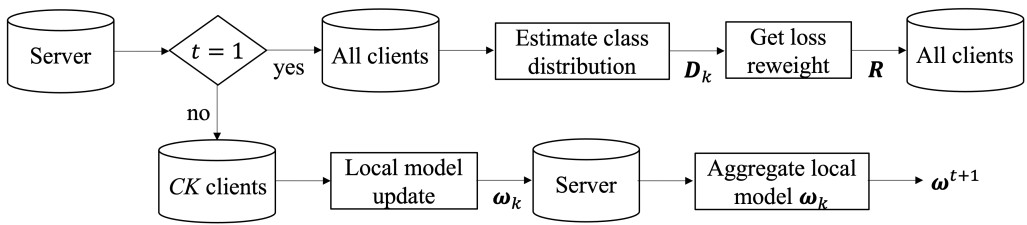

Figure 5: Flow chart of proposed method in each iteration. Note that in $t = 1$ only the class distribution estimation is performed. The global model is not updated in first iteration.

---

**Algorithm 2** FedRE: Fedrated learning with Ratio Estimation

---

**Input:** maximum global iteration $T$, local minibatch size $B$, number of local epochs $E$, learning rate $\eta$, clients index set $K$, weighted loss function $L$, balance auxiliary dataset $\mathcal{A}$
**Output:** model $\boldsymbol{\omega}^{T+1}$
1: initialize $\boldsymbol{\omega}^0$
2: **for** $t = 1, 2, ..., T$ **do**
3:     **if** $t = 1$ **then**
4:         **for** all $k \in K$ **do in parallel**
5:             $\boldsymbol{\omega}_k^E \leftarrow modelUpdate(\boldsymbol{\omega^0}, B, E, \tilde{\eta}, \mathcal{D}_k, l)$            ▷ line 2 in Algorithm 1
6:             upload $\boldsymbol{\omega}^E$ to server
7:         **end for**
8:         estimate $\hat{\boldsymbol{D}}_k$ for all $k$ by $\hat{\boldsymbol{D}}_k \leftarrow \frac{1}{|\mathcal{A}|} \sum_{i=1}^{|\mathcal{A}|} f(\boldsymbol{\omega_k^E}, \boldsymbol{x}_i)$     ▷ line 3 in Algorithm 1
9:         $\boldsymbol{R} \leftarrow getLossReweight([\hat{\boldsymbol{D}}_1, ..., \hat{\boldsymbol{D}}_K], [n_1, ..., n_k])$
10:        Send $\boldsymbol{R}$ to all clients
11:         **continue**
12:     **else**
13:         randomly sample $CK$ clients and create selection set $\mathcal{S}_t$
14:         **for** $k \in \mathcal{S}_t$ **do in parallel**.
15:             $\boldsymbol{\omega}_k^{t+1} \leftarrow modelUpdate(\boldsymbol{\omega}^t, B, E, \eta, \mathcal{D}_k, L, \boldsymbol{R})$    ▷ Clients local update by SGD
16:         **end for**
17:         $\boldsymbol{\omega}^{t+1} \leftarrow \sum_{k=1}^{|\mathcal{S}_t|} \frac{n_k}{N} \boldsymbol{\omega}_k^{t+1}$                 ▷ Aggregation step
18:     **end if**
19: **end for**

**Function** $getLossReweight([\hat{\boldsymbol{D}}_1, ..., \hat{\boldsymbol{D}}_K], [n_1, ..., n_k])$:
1:  $\hat{\boldsymbol{D}}_g \leftarrow \sum_{k=1}^{K} \frac{n_k}{n} \hat{\boldsymbol{D}}_k$
2:  $\boldsymbol{R} \leftarrow \alpha + \frac{\beta}{\hat{\boldsymbol{D}}_g^2}$
3:  **return** $\boldsymbol{R}$

---

# 4 EXPERIMENTAL RESULTS

The effectiveness of the proposed method are verified by several experiments, and the performance are compared with other FL algorithms. The experimental results highlight the advantages of the proposed method in the presence of severe heterogeneity on class distribution and mismatch between local and global imbalance. Three other algorithms, FedAvg (McMahan et al., 2017), FedNova (Wang et al., 2020b), and Ratio-Loss (Wang et al., 2021) are used in the comparison. FedAvg is the most widely-used base-line algorithm in FL. FedNova is one of the most common variation of FedAvg, which applies a normalized averaging method to eliminate the effect of data heterogeneity. Ratio-Loss utilizes a monitoring scheme to estimate the class distribution and detect the presence of global class imbalance, and then applies a novel loss re-weight method to mitigate the effect of class imbalance.

## 4.1 Experiment Setup

All the algorithms are implemented in PyTorch, and methods from other works are reimplemented in the experiments. Two different datasets: MNIST and CIFAR10 are chosen, which are very common benchmark dataset and frequently used for verification of FL algorithms. Different models are used for the tests on different datasets. To have a fair comparison, for each dataset the same model will be used for comparing different FL algorithms. The model used in MNIST dataset consists of one convolutional layer having 128 $3 \times 3$ filters with ReLU activation followed by 2 fully connected layers having 1000 and 100 neurons with Sigmoid activation. The model used in CIFAR10 dataset consists of 2 convolutional layers having 128 $5 \times 5$ filters with ReLU activation followed by 2 fully connected layers having 394 and 192 neurons with Sigmoid activation. For all clients, in estimation step, the Square Error Loss is used and the learning rate $\tilde{\eta} = 0.01$. In local update, the weighted Cross Entropy Loss stated in Eq.(2) is used and the learning rate $\eta = 0.05$. The minibatch size $B = 32$, the local training epoch $E = 5$, and the local optimizer is SGD in both estimation and local update.

To simulate the presence of globally imbalance class distribution, we simulate two kinds of scenarios that there are 1 and 2 minority classes within all classes, respectively. Each minority class has in total $s$ samples, and for the reset of the classes, each of them has $\rho s$ samples, where $\rho$ is the level of imbalance. In the experiment, to test the validity of the algorithm, two levels of imbalance $\rho = 5$ and $\rho = 10$ are chosen, and $s = 300$ and $s = 150$, respectively. In other words, there are 1500 samples belongs to each non-minority classes for both two imbalance level. These globally imbalanced samples are randomly distributed into each client by Dirichlet distribution to simulate the heterogeneity of local client data distribution, which also simulate the presence of mismatch between local and global class distribution.

For the Federated Learning environmental setting, two kinds of selection scenarios are built, which are the picking 5 clients out of 5 ($C = 1$) and picking 5 clients out of 16 ($C = 0.3$), respectively. For experiments on MNIST and CIFAR10, the global iteration numbers are 100 and 150, respectively. In the server, there is a balanced testing dataset consists of 32 samples for each class. The balanced auxiliary dataset for distribution estimation also consists of 32 samples for each class, in total 320 samples. Note that compared to the size of training set, the auxiliary dataset is much smaller, which incurs not much computational cost.

## 4.2 Results Summary

The experimental results of classification accuracy for 4 different methods with 1 and 2 minority classes are summarized and compared in Table 2 and Table 3. The numbers shown in the table without parentheses are overall classification accuracy, and the numbers with parentheses are classification accuracy of minority class. Note that for the case with 2 minority classes, the worst classification accuracy on 2 minority classes is reported in order to provide a fair comparison. As one can see in the tables, in both sets of experiments with different number of minority classes and different level of heterogeneity, the proposed method FedRE outperforms all other methods in terms of both overall and minority class classification accuracy. All other methods fail in correctly classifying the minority class, even if they still have acceptable overall accuracy. The overall performance of FedNova is very similar to the baseline method FedAvg in the experiments. This is because even thought FedNova can handle client heterogeneity, it is focusing on the data heterogeneity, that is, the difference in quantities of samples for each clients. Thus, it has minor effect on resolving the heterogeneity of imbalance class distribution. Ratio-Loss is shown to have a better performance than FedAvg and FedNova in general. However, it still can't tackle the issue with highly imbalance distribution in the experiments. On the other hand, the proposed method successfully achieves the bast performance in overall accuracy and classification accuracy on minority class in different level of heterogeneity. In the experiments, especially in tests on MNIST dataset, one can observe that the overall accuracy is boosted significantly. One can also observe that the overall accuracy is significantly improved in the experiments with 2 minority classes. These improvements are mainly because of the success in tackling imbalance class distribution issue. Other methods which fail in tackling the issue are affected by majority classes and end up with not learning the minority class. According to these experimental results, we can conclude that the proposed algorithm works effectively and provide a significant improvement on minority class classification accuracy without compromising the overall accuracy.

| | $\rho$ | $C$ | FedAvg | FedNova | Ratio-Loss | FedRE |
|---|---|---|---|---|---|---|
| MNIST | 5 | 1 | 88.13% (0.0%) | 88.13% (0.0%) | 88.75% (0.0%) | **98.44%** **(93.75%)** |
| | | 0.3 | 80.31% (0.0%) | 80.94% (0.0%) | 88.13% (0.0%) | **96.88%** **(93.75%)** |
| | 10 | 1 | 88.44% (0.0%) | 88.44% (0.0%) | 89.38% (0.0%) | **98.13%** **(93.75%)** |
| | | 0.3 | 81.25% (0.0%) | 80.36% (0.0%) | 88.44% (0.0%) | **97.5%** **(93.75%)** |
| CIFAR10 | 5 | 1 | 51.25% (0.0%) | 51.88% (0.0%) | 54.69% (0.0%) | **55.00%** **(40.63%)** |
| | | 0.3 | 37.81% (0.0%) | 37.5% (0.0%) | 40.63% (0.0%) | **48.44%** **(62.5%)** |
| | 10 | 1 | 50.31% (0.0%) | 50.94% (0.0%) | 53.13% (0.0%) | **54.06%** **(37.5%)** |
| | | 0.3 | 33.13% (0.0%) | 34.06% (0.0%) | 42.81% (0.0%) | **44.69%** **(40.63%)** |

Table 2: Performance comparison between different algorithms with 1 minority class. Results with respect to different datasets, level of imbalance $\rho$, and client selection ratio $C$ are reported. The numbers without parentheses are overall classification accuracy and numbers with parentheses are minority class classification accuracy.

| | $\rho$ | $C$ | FedAvg | FedNova | Ratio-Loss | FedRE |
|---|---|---|---|---|---|---|
| MNIST | 5 | 1 | 79.06% (0.0%) | 79.38% (0.0%) | 79.38% (0.0%) | **98.13%** **(93.75%)** |
| | | 0.3 | 71.88% (0.0%) | 72.19% (0.0%) | 78.44% (0.0%) | **96.56%** **(90.63%)** |
| | 10 | 1 | 78.44% (0.0%) | 78.44% (0.0%) | 79.69% (0.0%) | **97.81%** **(87.5%)** |
| | | 0.3 | 70.94% (0.0%) | 70.94 (0.0%) | 76.88% (0.0%) | **96.25%** **(87.5%)** |
| CIFAR10 | 5 | 1 | 49.06% (0.0%) | 48.75% (0.0%) | 50.94% (0.0%) | **53.44%** **(9.38%)** |
| | | 0.3 | 26.25% (0.0%) | 30.31% (0.0%) | 39.38% (0.0%) | **45.63%** **(34.38%)** |
| | 10 | 1 | 50.63% (0.0%) | 50.63% (0.0%) | 52.5% (0.0%) | **55.13%** **(3.13%)** |
| | | 0.3 | 25.62% (0.0%) | 30.0% (0.0%) | 39.06% (0.0%) | **45.31%** **(6.25%)** |

Table 3: Performance comparison between different algorithms with 2 minority classes. Results with respect to different datasets, level of imbalance $\rho$, and client selection ratio $C$ are reported. The numbers without parentheses are overall classification accuracy and numbers with parentheses are the worst minority class classification accuracy.

## 5 CONCLUSION

In this paper, a new algorithm based on a novel class distribution estimation method is proposed to tackle the class imbalance issue in Federated Learning with the presence of global class imbalance and distribution mismatch between local and global dataset. A class distribution estimation method is developed based on the experimental observation, in which the model parameters converge to a local minimum in the early stage of training where the prediction outputs are equal to the class distribution ratios. The class distribution estimation method is verified with both experiments and theoretical analysis. With the developed estimation method, the proposed algorithm utilizes the estimated information to perform loss re-weight to tackle the class imbalance issue. The validity and the effectiveness of the proposed algorithm are verified in extensive experiments with a variety of practical settings in FL, which also shows the superiority of the proposed method over other arts.

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
