# OpenReview forum: "Tackling Imbalanced Class in Federated Learning via Class Distribution Estimation"
_ICLR.cc/2023/Conference — Submitted to ICLR 2023_

### Official Review · Reviewer_wyG4 · 2022-10-24

**Confidence:** 4
**Clarity, Quality, Novelty And Reproducibility:** The paper is overall clearly written.
**Correctness:** 3
**Technical Novelty And Significance:** 2
**Empirical Novelty And Significance:** 2
**Recommendation:** 5

**Strength And Weaknesses:**

Pros:

The method of class distribution estimation is novel, and the observation is very interesting.

Cons:

**1, Lack of empirical support for the observation.** While the class distribution method is novel, it is only motivated by toy examples. It would be great to see this trend on real datasets and large neural networks.

**2, The theorem is not clear.** The paper provides a theoretical result to support the estimation method. However, it is hard to directly see why this is the case from the description of the theorem alone.

Questions:

1, It is not clear why local loss functions are reweighted with global class ratios. Shouldn’t each client have its own imbalance ratios?


**Summary Of The Paper:**

The paper tackles the global class imbalance problem in Federated Learning and proposes a novel class distribution estimation method. The estimation method is motivated by the observation that the averaged output probability vector from a classifier at the beginning of training is numerically close to the class ratio. The averaged class ratio is then integrated into the local model training in the form of weighted loss.


**Summary Of The Review:**

While the reviewer appreciates the insights and novelty of the class estimation method, it is poorly supported empirically and theoretically. More experiments on real datasets and larger neural networks, and better description and discussion of the theorem should help improve clarity and make the motivation more convincing.

---

### Official Review · Reviewer_8crs · 2022-10-28

**Confidence:** 4
**Correctness:** 2
**Technical Novelty And Significance:** 3
**Empirical Novelty And Significance:** 2
**Recommendation:** 3

**Clarity, Quality, Novelty And Reproducibility:**

clarity: writing is easy to follow.

quality: the mechanism needs auxiliary data, which makes the practical usefulness limited.

novelty: the class distribution estimation method seems to be new, but the theoretical assumption is too limited for practical data.

**Strength And Weaknesses:**

S1. The paper is easy to follow.

S2. Some theoretical analysis is provided.

W1. The assumption of theorem 1 is unrealistic (i.e., the input distribution of all classes is equivalent). While the authors provide an example of Fig. 4 to discuss that this can be addressed with the increase of network layers, it is insufficient. More experiments on different types of real data are needed for empirical analysis, and also a theoretical foundation should be established.

W2. The authors’ method needs balanced auxiliary data A for the server to estimate class distribution for each client. In practice, a high-quality A is very hard to obtain (how about the feature distribution of A being different from clients’ data?). Getting high-quality A is itself a very challenging problem in FL. Not to say, for many scenarios, no data can be collected on the server (this is why FL is needed). Then, the usefulness of the authors’ mechanism is doubtful.


**Summary Of The Paper:**

This paper proposes a privacy-preserving class distribution estimation method for dealing with class-imbalanced FL scenarios.

**Summary Of The Review:**

While the class distribution estimation method is interesting, it needs more theoretical or empirical justifications. Moreover, it needs to carefully find a scenario where high-quality data A can be obtained; otherwise, the usefulness of the mechanism is very limited.

---

### Official Review · Reviewer_oRX1 · 2022-10-29

**Confidence:** 4
**Correctness:** 3
**Technical Novelty And Significance:** 2
**Empirical Novelty And Significance:** 2
**Recommendation:** 3

**Clarity, Quality, Novelty And Reproducibility:**

The paper is more or less clearly written, although the linguistic quality could improve,

The novelty is moderate as global class reweighting for imbalanced classification task has been used frequently before in mainstream machine learning.

Technical quality and experimental validation part is poor as I already elaborated before. Consequantly the reproducibility is also not up to the expected level.

**Strength And Weaknesses:**

Unfortunately, I noted a number of issues regarding the technical quality of the paper and these are elaborated below.

1) Paragraph below Theorem 1 (page 5) states that the noise arising out of any violation of above assumption can be eliminated by adding more number of layers. This is demonstrated by some results in fig 4, but no proof or theoretical discussion has been provided.

2) In results, accuracy for minority class in all SOTA algorithms are shown to be ABSOLUTELY zero, whereas the proposed algorithm reaches considerable accuracy (except for a few rows for CIFAR10 dataset where it is single digit). It should be investigated if they have implemented the SOTA algorithms correctly.

3) The brief survey on methods for handling class imbalance in the Introduction section is a bit outdated. Given the problem is quite popular there are many majors developments that took place in recent years. Can you please explain a bit more on why data level imbalance handling techniques cannot be applied locally? If possible can you please cite a reference as you did for the case of cost sensitive learning.

4) While it is somewhat clear to me what the paper is aiming to propose the motivation remains a bit vague. A bit more clarity on motivation and a glimpse of the algorithm preferably with an illustrative example will only help to highlight the contributions and novelty.

5) The simulation study I felt is too simple. A more complex example may be using an imbalanced version of MNIST or Fashion-MNIST using a deeper CNN and a high imbalance ratio may be more beneficial.

6) Section 3.2 introduces three new hyper-parameters. Unfortunately,  we do not know how the plateau region behaves with the three hyper-parameters or if its behaviour changes in any way in practice. Also I did not understand how the other cited stopping criteria can aid us or why they are mentioned if not in the scope of the current article.

7) The getLossReweight is a key ingredient of the proposed algorithm and deserves its own place in the paper. Also no intuitive explanation behind the heuristic of converting the estimated priors to corresponding weights is given. Is there any ablation study on this using different choices of functions?

8) Why Accuracy is used in an imbalanced setting given we know it fails to properly evaluate the performance is such cases? Especially, for Table 3 we have only the worst performing minority class performance to compare on. That Minority class may change over algorithms and if that happens the comparison may become unfair to an extent. Also what happens to the second minority class remains a mystery. Can you use some indices that provides an unbiased evaluation of the classifier in presence of class imbalance (see for example, https://doi.org/10.1016/j.patcog.2020.107197)? I also felt the experimental setup is somewhat simple. May be you can order the classes and select different number of samples from each class, so that 1. Pairwise imbalance ratios will vary and 2. The overall imbalance ratio can be made really high like in the range of 50-100.You can also check how the estimated class priors match to that of the real one.

9) One problem of cost sensitive learning is that it can overcompensate. In effect this may bring down the TPRs for relatively majority classes. Can you please confirm if such a case is happening?

10) The algorithm assumes we can generate synthetic data that is balanced, or some real-life balanced dataset is publicly available to server. Balanced auxiliary dataset is mentioned as input to Algorithm 2 and used in line 8 of that algorithm. How much practically relevant is that?



**Summary Of The Paper:**

The paper presents a class-distribution estimation based framework for federated learning. The method incorporates loss reweighting scheme for handling global class imbalance. The paper is completely based on the assumption that after certain rounds of training with imbalanced class, the class probability returned by any sample will be equal to the global distribution of the classes. E.g. if global distribution of a 3-class problem is 5%, 10% and 85%, the returned softmax for each sample will be close to 0.05, 0.10, and 0.85. This is shown with a 2-class example experiment (page 4) and stated as Theorem 1 (page 5).

**Summary Of The Review:**

The paper addresses an important issue in federated machine learning. However, the current version is far from being matched to the high standards expected for a conference like ICLR.

---

### Decision · Program_Chairs · 2023-01-20

**Decision:**

Reject

**Justification For Why Not Higher Score:**

No author response was provided.

**Justification For Why Not Lower Score:**

No author response was provided.

**Metareview: Summary, Strengths And Weaknesses:**

The reviewers had some major concerns which were not alleviated since no author response was provided. Hopefully, in the revised version of the paper the concerns of the reviewers will be addressed.